# Unpacking the Suitcase
# of Semantic Similarity

## Abstract

Retrieval-augmented generation (RAG) has become a *de facto* standard for reducing factual inaccuracies in LLM-generated responses and it is generally accepted that cosine similarity between two text embeddings is a state-of-the-art measure of semantic similarity. In practice, however, there is a disconnect between the expectation to retrieve semantically highly relevant text and the kinds of information text embeddings actually represent. The aim of this study is to unpack the generic term "semantic similarity" into empirically distinguishable components and investigate how they factor into the cosine similarity of text embeddings. We derive analytic expressions for semantic entailment similarity on concept, predicate and proposition levels based on a previously proposed logical framework of conceptual semantics. This enables us to create a benchmark dataset of concepts and propositions with quantitatively characterized semantic entailment relationships. We train linear projections from the text embeddings of 15 state-of-the-art embedding models to semantic entailment space and assess the deviation of semantic entailment cosine similarity estimates from the ground truth. Next, we identify proposition entailment similarity categories that are relatively more difficult to handle for low than high performing models. As a complementary approach, regression modeling is used to demonstrate the predictive value of symbolic similarity, contextual similarity and entailment similarity on the cosine similarity of text embeddings. Both approaches are found to converge on a small set of models that are significantly better at semantic entailment estimation than the rest. We conclude that the majority of variation in cosine similarity of text embeddings is due to contextual similarity as opposed to entailment, and propose using the term "**contextual similarity**" instead of the ambiguous "semantic similarity" when referring to cosine similarity estimates from text embedding models. We also propose the term "**contextual fingerprint**" to capture the intuition behind text embeddings instead of the potentially misleading "semantic embedding".

## 1 Introduction

> *"I personally was never interested in the so-called problem of meaning; on the contrary, it appeared to me a verbal problem, a typical pseudo-problem."*
>
> — Karl Popper, *Conjectures and Refutations*

Retrieval-augmented generation (RAG) has become a *de facto* standard for reducing factual inaccuracies in responses from large language models (LLMs) Lewis et al. (2020). However, the semantic retrieval mechanisms based on text embeddings tend to be noisy and benefit substantially from additional filtering by re-ranking models and LLMs Xu et al. (2025); Feng et al. (2021). Nevertheless, the cosine similarity between two text embeddings from dedicated models is generally considered to be a state-of-the-art measure for text retrieval based on semantic similarity Reimers & Gurevych (2019); Guo et al. (2021). This leads us to suspect that, at least among the users of the RAG approach, there is often a palpable disconnect between the expectation to retrieve highly relevant information on the level of semantic entailment and the kinds of information actually represented in text embeddings.

While it might not be surprising that semantic similarity is an overloaded "suitcase" term with many meanings, even semantic entailment can be difficult to pin down Lee et al. (2025). In logic, en-

tailment refers to the situation where a statement necessarily follows from the premises. Textual or natural language entailment carries a more relaxed definition whereby a statement $P$ entails a statement $h$ if a human reading $P$ would infer that $h$ is most likely true Bos & Markert (2005); Putra et al. (2024). As such, natural language entailment assumes a shared world model between the sender and receiver implying that an unspecified amount of contextual information is needed to decide whether the entailment relation holds between $P$ and $h$ (as in e.g., Romanov & Shivade (2018)). Lexical entailment, on the other hand, has been defined as a semantic relation between verbs where $V_1 \implies V_2$. This is exemplified by *"if the subject snores then the subject also sleeps"* leading to *snores $\implies$ sleeps* Fellbaum (1990).

Here, we will be addressing yet another form of semantic entailment where the entailment between conceptual entities is decided based on the overlap of their referents and the entailment between predicates is decided based on the stipulation of logical implication or contradiction between them in an otherwise fixed propositional context. Specifically, we are interested in quantifying the amount of semantic entailment between two linguistic expressions that map either to concepts with enumerable sets of referents or to atomic propositions with an atomic predicate that accepts concepts of the former kind as arguments.

Based on the conceptual semantics framework proposed in a concurrent submission, we are going to derive analytic expressions for the quantification of semantic entailment similarity on concept, predicate and proposition levels that can reflect logical implication, contradiction and independence. We create a benchmark dataset of concept labels and sentence pairs constructed according to predefined formulas based on a restricted vocabulary of concept and predicate labels with well-defined semantic entailment relationships. The proposed semantic entailment measure is used to obtain the ground truth estimates of entailment similarity in the dataset. Based on this dataset, we train linear projections on the outputs of text embedding models to estimate semantic entailment via cosine similarity. We rank 15 state-of-the-art text embedding models based on correlation with the ground truth and identify the statistically outstanding ones. Finally, we create a trial dataset highlighting the tension between contextual and entailment similarities in sentence similarity tasks. We use linear and generalized linear regression to estimate the contribution of various aspects of sentence similarity to cosine similarities produced by the text embedding models on the trial dataset.

## 2 RELATED WORK

While datasets devoted to sentence-to-sentence semantic similarity Marelli et al. (2014); Khot et al. (2018); Dalvi et al. (2021) create opportunities for training predictors, they will not reveal the contribution of various flavors of sentence similarity on the level of text embeddings unless designed with that objective in mind. A recent review on the explainability of text embeddings Zhang et al. (2024) noted the sparseness of relevant studies and concluded that there is a significant gap in the literature regarding the direct examination of the interpretability and explainability of word embeddings. Numerous studies have focused on decomposing standard embeddings into orthogonal sub-spaces that represent specific linguistic features like polarity, concreteness, part-of-speech, and word sense while semantic entailment is not always included Opitz & Frank (2022); Sun & Platoš (2023); Yaghoobzadeh et al. (2019); Jang & Myaeng (2017).

Text embeddings from language models have been used for information retrieval via semantic search Reimers & Gurevych (2019); Zhu et al. (2023); Guo et al. (2021); Lin et al. (2021), to increase the explainability of reasoning during question answering Wang et al. (2024), to improve uncertainty quantification in LLMs Grewal et al. (2024) and a host of other semantically motivated tasks. Semantic retrieval combined with a generative language model corresponds to retrieval-augmented generation Lewis et al. (2020) that is a popular technique for anchoring query-focused responses from the language model to a curated knowledge base. The contribution of a generative language model in RAG is two-fold. First, it acts as a semantic filter that locates relevant information in the retrieved context while ignoring the irrelevant. This is the reason why retrieval systems with dubious specificity are still useful as long as the recall rate is adequate. Secondly, a generative model can phrase a response that is tailored to the semantic and formatting expectations provided in the query. For improved retrieval, re-ranking models Xu et al. (2025); Feng et al. (2021) have been incorporated into the RAG pipeline to increase retrieval specificity. More specific context will increase the

relevance of responses since generative language models tend to respond in overly general terms when query-specific information is not available.

Faithfulness of LLM-generated text to factual statements is becoming increasingly important as LLMs are becoming routinely employed as assistants in knowledge work. Cattan et al. (2024), for example, use decomposing of LLM-generated text into minimal predicate-argument level propositions to identify whether the generated statement is entailed by the supporting context. Using LLMs to resolve semantic entailment between two propositions is definitely an effective solution, but it is not easily scalable to billions of statements. Accordingly, effective representation of entailment similarity on embedding level would ultimately present a more cost-effective solution.

## 3 METHODS

Please see Appendix A for the derivation of semantic entailment similarity measures.

### 3.1 CONSTRUCTION OF RESTRICTED VOCABULARY

We sourced 838 synonymous label pairs for conceptual entities from the LLM responses in the "limited-list-referents" task from the conceptual integrity benchmarking study that is being submitted concurrently (referencing is currently not possible due to limitations imposed by the double-blind review process). Synonym pairs were validated using GPT-4.1 (see prompt template in Appendix D.1) as being reasonably common and, if possible, unambiguously associated with the same conceptual entity across various contexts. A pair of synonyms is connected via a bidirectional semantic entailment relationship as they refer to the same entity.

Concept labels from parent (category label) and child (category members) entities were harvested from all concepts with a nested set of referents in the dataset provided in the conceptual integrity benchmarking study to yield 7436 pairs of parent-child relationships with ground truth semantic entailment values. These concept labels originated from well-known scientific ontologies used routinely by researchers. Parent-child concepts are connected via one-way implication: using the parent term implies (refers to) all of its children while a single child term implies only itself and not any of its siblings. See Appendix C.2 for examples.

Conceptual relations (predicates) were harvested from ten thousand biomedical research publication abstracts downloaded from the PubMed National Center for Biotechnology Information (2025) database. The language model (GPT-4o-mini) was instructed to extract predicates with associated subject and object from the sentences referring to the findings of the study (see prompt template in Appendix D.2). Next, the predicates were clustered into relational domains with help from the LLM (GPT-4.1, see prompt template in Appendix D.3). The clusters were then merged into non-redundant relational domains using an agentic script that asked GPT-4.1 to iterate through all clusters and decide whether the current cluster can be combined with any of the clusters processed so far or not (prompt template in Appendix D.5). Any remaining inconsistencies and redundancies were resolved by a human annotator to yield 37 relational domains covering 246 predicates related to expressing research findings in biomedical publications. Human annotated set was re-checked for consistency using the prompt template Appendix D.6 and it was up to the annotator to accept or decline suggestions from the LLM. See Appendix C.1 for examples of relational domain definitions.

### 3.2 SAMPLING OF CONCEPT AND PROPOSITION PAIRS WITH WELL-DEFINED SEMANTIC ENTAILMENT RELATIONSHIPS

For the concepts dataset, 2500 concept pairs were sampled from the restricted vocabulary with the following semantic entailment relationships: equivalent conceptual entities (416), parent-child conceptual entities (417), unrelated conceptual entities (417), equivalent predicates (417), contradictory predicates (403), unrelated predicates (430).

The propositions dataset was created (prompt template in Appendix D.4) by generating 1500 propositions according to the $C_1 \times C_2 \times R$ construction formula where $C_1$ and $C_2$ are conceptual entities (numbered in the order of occurrence in the sentence) and $R$ is a conceptual relation. In the samples, $C_1$ and $C_2$ predominantly serve the roles of the grammatical subject and object, but there are numerous exceptions easily recognized by those skilled in the art.

For example, the sentence "Chlorine dioxide increased rhamnose levels in bacteria" maps to the construction formula $[C_1 = $ "Chlorine dioxide"$][R = $ "increased"$][C_2 = $ "rhamnose"$]$ "levels in bacteria".

The context for the subject and object in the proposition was created by randomly sampling 3 candidate concept pairs with well-defined semantic entailment relationships (see Appendix E for details). As candidates for the predicate, 3 relational domains were sampled. Next, a language model (GPT-4.1) was tasked with constructing two contextually plausible sentences satisfying a predefined semantic entailment relationship by picking corresponding terms from the available candidates. The instructions required that semantically related sentences differ only in terms of the $C_1$, $C_2$ and $R$ components. The language model was allowed to decline and request a new context if the candidates did not yield plausible sentences. The resulting propositions were annotated for symbolic and semantic entailment similarity and checked for validity using deterministic algorithms.

Character level similarity was calculated using Jaro-Winkler similarity algorithm implemented in Python's *jarowinkler* package. Token similarity was calculated as the average of Jaccard similarity (ratio between the number of intersecting tokens to unique tokens in two sequences) and a variant of Ratcliff-Obershelp sequence similarity measure implemented in Python's *difflib* package.

A total of 1500 propositions was constructed with 100 propositions for each predefined category of semantic entailment similarity except for the category 2.3 "one-way implication between predicates". Since we were not aware of a suitable knowledge base of predicate hierarchies and the semantic entailment dataset was already sufficiently large and novel, we decided to leave it for future investigation.

### 3.3 Selection of embedding models for benchmarking

Fifteen state-of-the-art embedding models were picked from the global ranking in the MTEB Leaderboard Muennighoff et al. (2022) with ranks ranging from 2 to 199 while the majority (10) of the models ranked within the top 30 (as of 10 August 2025). Commercial models were used via internalized APIs while the open source models were downloaded from the HuggingFace model repository Hugging Face (2023).

### 3.4 Construction of trial dataset

A trial dataset of 16 examples was constructed to study the predictive value of semantic entailment and contextual similarities when estimating sentence-to-sentence cosine similarity from text embeddings. The dataset was designed to be complete in terms of representing the full scale of semantic entailment similarity values ($a \implies b \in [-1, 0, 1]$) in all possible combinations with the contextual switching flag (Table 1). Furthermore, all combinations were represented uniformly in order to obtain unbiased estimates.

Table 1: Trial dataset composition

| Context Switch | Entailment | Description | Number of examples |
|---|---|---|---|
| 0 | 1 | equivalence | 4 |
| 0 | 0 | independence | 4 |
| 0 | -1 | contradiction | 4 |
| 1* | 0 | independence | 4 |

*Entailment is always zero for sentences that involve a major shift in context.

### 3.5 Training of linear projections

Linear projections from the model's native embedding dimensions to the following output dimensions were trained: 50, 100, 200, 400, 800, 1600 and 3200. Linear projections were chosen to introduce a minimal amount of auxiliary information into the text embeddings. Multiple output

dimensions were used to ensure repeatability and generalization of results across various output dimensions. Weights were initialized to a gaussian distribution with mean 0 and standard deviation 0.02. Learning rate was set to decay from 1e-4 to 1e-5 over 200 iterations. Projections of all sizes were trained with AdamW optimizer Loshchilov & Hutter (2019) for 200 iterations (batch size 32) on 80% of the training data while 20% of the data was used to monitor overfitting. No early stopping and weight decay was used. The training objective was to reduce the mean square error between the cosine similarity of projected embeddings and the ground truth estimate. The training loss is given by the formula:

$$\mathcal{L}(W) = \frac{1}{N} \sum_{i=1}^{N} \left( \text{cosine\_sim}(a_i W, b_i W) - y_i \right)^2 \tag{1}$$

$$= \frac{1}{N} \sum_{i=1}^{N} \left( \frac{(a_i W) \cdot (b_i W)}{\|(a_i W)\|_2 \cdot \|(b_i W)\|_2} - y_i \right)^2 \tag{2}$$

where $a_i, b_i \in \mathbb{R}^{1 \times d_{\text{in}}}$ are the original text embeddings, $W \in \mathbb{R}^{d_{\text{in}} \times d_{\text{out}}}$ is the projection matrix to be learned, $y_i \in \mathbb{R}$ is the ground truth semantic entailment similarity, $N$ is the number of training pairs, and $\| \cdot \|_2$ denotes the L2 norm.

For each model, separate projections were trained for four distinct endpoints:

1. Semantic similarity endpoints
   (a) $a \implies b$ (semantic entailment similarity from $a$ to $b$)
   (b) $b \implies a$ (semantic entailment similarity from $b$ to $a$)
2. Symbolic similarity endpoints
   (a) $char(a \sim b)$ (character-level similarity)
   (b) $token(a \sim b)$ (token-level similarity)

### 3.6 RANKING SEMANTIC ENTAILMENT ESTIMATION PERFORMANCE IN EMBEDDING MODELS

Since ranking by Pearson and Spearman correlations yielded statistically very similar results, the ranks were called based on Pearson product moment correlation. Correlation-based rankings were obtained for all four endpoints. To identify models that consistently ranked among the top in semantic or symbolic endpoints, rankings from two related endpoints were pooled. To get a statistical estimate of a model consistently ranking to the top, we recorded the number of times ($k$) a model ranked among the top 5 out of 15 benchmarked models. The cumulative probability of making it into the top third ($p = 1/3$) on $k$ or more occasions out of $n = n_{dataset} \times n_{relevant\_endpoint} \times n_{out} = 2 \times 2 \times 7 = 28$ attempts was estimated based on the binomial distribution. P-values were corrected for multiple testing using the Benjamini-Yekutieli procedure Benjamini & Yekutieli (2001) to obtain the q-values. Models with $q < 0.05$ were considered significantly enriched among the top performers.

### 3.7 RANKING CATEGORIES OF SEMANTIC ENTAILMENT RELATIONSHIPS BETWEEN PROPOSITIONS

Our aim was to reveal what types of proposition pairs exhibited largest differences in semantic entailment estimation error between high vs low performing models. Pairs of compared models are found in Appendix Table 10. We calculated the absolute difference between the predicted semantic entailment similarity from projected embeddings and the ground truth value for endpoint $a \implies b$ for each sentence pair in the propositions dataset.

Estimation errors were grouped by projection's output size and entailment category followed by the application of the Wilcoxon signed rank test (a paired test) to identify the significance of the difference in errors between two models in each category. Categories were ranked by the descending differential of error $e_{c_i}^B / e_{c_i}^A$ where $e$ is the average absolute prediction error for category $c_i$ in the corresponding model ($A$ or $B$). From the nine model-to-model comparisons we

got $n = n_{model\_pair} \times n_{out} = 9 \times 7 = 63$ rankings of proposition categories based on descending differential of error between the high and low performing model.

## 3.8 REGRESSION ANALYSIS

In preliminary tests, it was found that using an absolute value of the semantic entailment measure $a \implies b$ significantly increased (t=-5.98, p<0.001, paired t-test) the predictability of sentence cosine similarity estimates from text embedding models ($R^2_{signed} = 0.865 \pm 0.0558$ vs $R^2_{absolute} = 0.889 \pm 0.057$). Accordingly, absolute semantic entailment was used as a predictor in linear and generalized linear regression analyses in order not to underestimate its contribution to sentence-to-sentence cosine similarity. To increase the comparability of regression coefficients between predictors, the data from each predictor was normalized to the zero mean and unit standard deviation. For generalized linear regression, second degree polynomial features were added (i.e. pairwise interactions and square terms) followed by the removal of highly correlated features to enhance the stability and interpretability of coefficients.

## 3.9 USAGE OF LLM ASSISTANTS IN MANUSCRIPT PREPARATION

GitHub Copilot was used to speed up the coding process, help to format the LaTeX document and to retrieve references to potentially related work via arXiv and Google Scholar search queries.

# 4 RESULTS

## 4.1 CONSTRUCTION OF THE DATASET OF SEMANTICALLY RELATED PROPOSITIONS

We constructed a restricted vocabulary of concept pairs from biology, chemistry and medicine exhibiting one or two-way semantic entailment relationships based on a dataset created in our previous work. Next, we downloaded 10,000 biomedical research abstracts from PubMed and used a language model to extract predicates that were used by researchers to express findings. The predicates were consolidated into 37 relational domains wherein predicates were related via entailment or contradiction. Each relational domain was partitioned into two or three subsets of predicates so that any pair of predicates was related via bi-implication within a subset and via contradiction between subsets (see Appendix C.1 for examples).

Based on the restricted vocabulary of concepts and predicates, we constructed two datasets of paired linguistic expressions ($a$ and $b$) with predefined semantic relationships. Four different semantic relationships were considered: (1) bidirectional entailment (two-way implication or equivalence), (2) asymmetric entailment (one-way implication), (3) contradiction, and (4) independence (i.e. absence of entailment and contradiction).

The datasets consisted of 2500 and 1500 pairs of linguistic expressions referring to conceptual entities and propositions, respectively. Examples of expressions referring to conceptual entities include "interleukin-1 receptor antagonist", "transaldolase" and "inflammatory bowel disease". Examples of conceptual relations (predicates) include "boosted", "was upregulated", "was genetically correlated with" etc. For each expression pair, two entailment similarity measures $a \implies b$ and $b \implies a$, and symbolic similarity measures on character-level $char(a \sim b)$ and token-level $token(a \sim b)$ were provided as the ground truth.

## 4.2 TESTING OF EMBEDDING MODELS

Our aim was to assess how accurately the output from state-of-the-art text embedding models was mappable to the semantic entailment measures via cosine similarity. The distribution of cosine similarities between paired expressions based on original embeddings ranged from 0.049 to 1 (with mean and standard deviation $0.701 \pm 0.2$) although cosine similarity also supports negative values. As the proposed semantic entailment measure ranges from -1 (contradiction) through 0 (independence) to 1 (entailment), it was deemed necessary to train a projection from the original embedding space to the semantic entailment space.

We chose to train linear projections of various output sizes (50, 100, 200, 400, 800, 1600, 3200) to introduce a minimal amount of auxiliary information into the embeddings, avoid biases from choosing a fixed output dimension and to ensure the repeatability of findings. It was reasoned that projections of embeddings with richer representation of semantic entailment information would align more closely with the theoretical estimates.

Separate projections were trained onto each ground truth similarity measure (endpoint) and models were ranked in the order of descending Pearson product moment correlation between the predicted and ground truth similarity estimates on each of the four endpoints. At least three embedding models were consistently ranking among the top 5 based on the accuracy of entailment similarity estimates (Table 2) as judged by the binomial probability distribution (see Methods section 3.6 for details). After repeating the training procedure three times it was found that e5-mistral-7b-instruct, text-embedding-3-large and gte-Qwen2-1.5B-instruct were in all runs significantly enriched among the top 5 performers in semantic entailment estimation tasks while bge-m3 and snowflake-arctic-embed-l-v2.0 stood out in symbolic similarity estimation rankings (see appendix F for overview).

Average correlation between ranking in semantic entailment estimation and in MTEB Leaderboard revealed a moderate and significant positive correlation (Spearman $\rho = 0.584$, $p = 0.0286$) indicating that performance on the semantic entailment measure proposed here agrees reasonably well with performance across established text embedding benchmarks. Small but insignificant negative correlation was observed between summarized rankings in the semantic and symbolic estimation tasks (Spearman $\rho = -0.182$, $p = 0.531$) indicating that the objectives are mostly orthogonal. Correlation of symbolic similarity measures with MTEB rankings was insignificant ($\rho = -0.252$, $p = 0.385$).

Table 2: Summary of rankings based on semantic entailment estimation accuracy

| Model | Sum of Ranks | Mean Rank | Times in Top 5 | p | q |
|---|---|---|---|---|---|
| **e5-mistral-7b-instruct** | 70 | 2.50 | 27 | 0 | **0** |
| **text-embedding-3-large** | 75 | 2.68 | 26 | $1 \times 10^{-10}$ | $\mathbf{2.49 \times 10^{-9}}$ |
| **gte-Qwen2-1.5B-instruct** | 128 | 4.57 | 18 | 0.001 | **0.013** |
| Qwen3-Embedding-8B | 141 | 5.04 | 15 | 0.022 | 0.268 |
| gte-Qwen2-7B-instruct | 181 | 6.46 | 14 | 0.050 | 0.501 |
| text-embedding-3-small | 197 | 7.04 | 12 | 0.191 | 1 |
| cohere.embed-multilingual-v3 | 206 | 7.36 | 9 | 0.623 | 1 |
| bilingual-embedding-large | 216 | 7.71 | 3 | 0.999 | 1 |
| bge-m3 | 219 | 7.82 | 10 | 0.464 | 1 |
| stella_en_1.5B_v5 | 228 | 8.14 | 4 | 0.994 | 1 |
| Qwen3-Embedding-0.6B | 311 | 11.1 | 0 | 1 | 1 |
| GIST-large-Embedding-v0 | 317 | 11.3 | 1 | 1 | 1 |
| snowflake-arctic-embed-l-v2.0 | 355 | 12.7 | 0 | 1 | 1 |
| gte-large-en-v1.5 | 357 | 12.8 | 1 | 1 | 1 |
| gte-base-en-v1.5 | 359 | 12.8 | 0 | 1 | 1 |

## 4.3 Dissecting task difficulty

Next, we compared errors of entailment similarity estimates from projected embeddings to understand what entailment categories (given in Appendix E) diverged the most in terms of accuracy between high vs low performers. To that end, we made pairwise comparisons between sibling models that differed in size (typically larger models performed better) and models that were top performers in entailment vs symbolic similarity estimation (see Appendix Table 10).

Entailment categories 1.2, 1.7, 3.1, 3.2 were significantly enriched at the top of differential error rankings and 1.4 came very close to significance (see Appendix Table 11). In general, the top half of the rankings was solely occupied by entailment categories from families 1 and 3 corresponding to various flavors of equivalent and contradictory propositions. This is somewhat surprising because the entailment family 2 that was represented in the lower half of ranks contains propositions with one-way implication which is identical to equivalence (category 1) in the $a \implies b$ direction (the

endpoint studied here) in the propositions dataset. Low error differential for family 2 suggests that entailment relationships between parent and child concepts are resolved with relatively similar accuracy between high and low performing models while entailment between synonyms (category 1) and contradictory predicates (category 3) is handled significantly better by high performers in the present dataset of biomedical and chemical entities.

In absolute terms, contradiction (family 3) and independence (family 4) were the most difficult entailment types to estimate for both high and low performing models based on the magnitude of the mean error. Relatively, however, high performing models were significantly better at estimating contradiction while errors in entailment estimates of independent propositions were more similar. The simplest entailment category (the one with lowest mean error) was clearly 1.3 where subjects and objects are identical between the two propositions and the predicates are synonymous. This finding suggests that identification of synonymous predicates is a task that both high and low performing embeddings models excel at.

## 4.4 Disentangling contextual and entailment similarity

We designed a small trial dataset that included pairs of sentences from various knowledge domains (biomedicine, astronomy, geography and arithmetic) with corresponding ground truth values for semantic entailment, character- and token-level similarity as well as a "context switch" flag. The context switch flag was set to 1 whenever the two sentences came from different knowledge domains and to 0 otherwise. Importantly, the dataset was balanced in terms of the number of examples corresponding to possible combinations of context switch and entailment values as indicated in Table 1.

The baseline linear model was constructed from the following predictors: character level similarity $char(a \sim b)$, token level similarity $token(a \sim b)$, absolute semantic entailment similarity $abs(a \implies b)$ and context switch $context(a \sim b)$. The generalized version of the model included the following predictors after the addition of second degree polynomial features and the removal of highly correlated ones: $abs(a \implies b)$, $token(a \sim b)$, $context(a \sim b)$, $abs(a \implies b)^2$, $token(a \sim b) \times context(a \sim b)$.

The mean ratio of sentence-to-sentence cosine similarity variation explained by the predictors ($R^2$) was $0.889 \pm 0.057$ vs $0.863 \pm 0.057$ (mean $\pm$ std) for the linear model with and without entailment similarity predictor. The same indicators for the generalized model were $0.906 \pm 0.048$ vs $0.869 \pm 0.0498$ indicating that the explainability of the cosine similarity measure increased by roughly 3% when including semantic entailment among the predictors.

The modest contribution of the semantic entailment predictor is mostly due to the large effect of context switching on cosine similarity. For example, according to the text-embedding-3-large model, the cosine similarity of sentences *"WFS1 protein deficiency activates XBP1."* and *"Moon orbits around the earth."* was 0.02 whereas the corresponding value for the pair *"WFS1 protein deficiency activates XBP1."* and *"WFS1 protein deficiency activates BNIP1."* was 0.8. On proposition level, semantic entailment is 0 in both cases (because *XBP1* and *BNIP1* are independent entities) whereby a large share of predictive value around the cosine similarity is carried by the context switch feature. This example clearly highlights that semantic entailment and contextual similarity operate on very different levels of semantic granularity.

In the generalized regression model, the relative size of the semantic entailment $abs(a \implies b)$ regression coefficient ranged between 1.42% to 20.51% of the sum of coefficients. Interestingly, when models were ranked by the descending relative size of the entailment coefficient (Table 3), there was a significant positive correlation with the semantic estimation accuracy ranking reported above that was based on projected embeddings ($\rho = 0.729$, $p = 0.002$). In other words, we were able to demonstrate on two different datasets and using two complementary approaches (linear projection of embeddings to ground truth entailment similarity vs generalized linear regression of ground truth similarity measures to cosine similarity of unprojected text embeddings) that text embeddings from specific models in our sample (e.g., e5-mistral-7b-instruct, text-embedding-3-large, gte-Qwen2-1.5B-instruct) appear to carry more semantic entailment information than others.

We believe that such convergence of results from different approaches indicates that the semantic entailment measure proposed here is a logically and empirically consistent phenomenon that is tractable in language models.

Table 3: Entailment coefficient size and model rankings

| Model | Entailment coefficient size (% of total) | Entailment accuracy rank | MTEB rank |
|---|---|---|---|
| e5-mistral-7b-instruct | 20.51 | 1 | 12 |
| text-embedding-3-large | 13.51 | 2 | 16 |
| Qwen3-Embedding-8B | 12.47 | 4 | 2 |
| cohere.embed-multilingual-v3 | 12.38 | 7 | 13 |
| gte-Qwen2-1.5B-instruct | 12.11 | 3 | 14 |
| Qwen3-Embedding-0.6B | 7.16 | 11 | 4 |
| bge-m3 | 7.14 | 9 | 22 |
| bilingual-embedding-large | 6.65 | 8 | 15 |
| snowflake-arctic-embed-l-v2.0 | 6.36 | 13 | 34 |
| text-embedding-3-small | 6.19 | 6 | 32 |
| gte-large-en-v1.5 | 5.75 | 14 | N/A |
| gte-Qwen2-7B-instruct | 5.01 | 5 | 6 |
| stella_en_1.5B_v5 | 2.87 | 10 | 18 |
| gte-base-en-v1.5 | 2.46 | 15 | 199 |
| GIST-large-Embedding-v0 | 1.42 | 12 | 55 |

## 5 DISCUSSION

Our results demonstrate that text embeddings from state-of-the-art embedding models primarily capture contextual similarity rather than semantic entailment relationships. We believe that this important insight is at the core of many frustrations when setting up a retrieval pipeline and it is often only partially addressed. The moderate contribution of semantic entailment to cosine similarity (approximately 3% improvement of $R^2$ in linear regression analysis) suggests that current embedding models are optimized more for topical and domain-level similarity than for entailment relationships between concepts and propositional constituents. This aligns with the principle of distributional semantics that "*a word is characterized by the company it keeps,*" which naturally leads to embeddings that cluster semantically related terms based on co-occurrence patterns rather than logical entailment.

The finding that better-performing models in semantic entailment estimation (e.g., e5-mistral-7b-instruct, text-embedding-3-large, gte-Qwen2-1.5B-instruct) show consistent superiority across the evaluation approaches used here validates our proposed semantic entailment measure as a meaningful and measurable property. The convergence of results from projected embeddings and regression analysis of cosine similarity provides evidence for the tractability of semantic entailment information in language models. Furthermore, we were able to break down proposition-level entailment into subcategories (Appendix E) which are differentially impacted in high vs low performers: while synonymous predicates are handled well by all models, contradictory relationships and equivalence between synonymous concept labels present greater challenges for lower-performing models. Since the predicates in the restricted vocabulary were most likely more common (i.e. less domain-specific) than the synonymous concept labels of genes, chemical compounds and diseases, this finding might be due to the specialized nature of our dataset. Nevertheless, these findings suggest that semantic nuance (involving various knowledge domains), rather than basic similarity detection, distinguishes top-performing embedding models in semantic entailment estimation.

In line with the aforementioned findings, we propose that the term "**contextual similarity**" be used in place of the ambiguous "semantic similarity" when referring to cosine similarity estimates obtained via text embedding models. Similarly, it would be pertinent to use the term "**contextual fingerprint**" instead of "semantic embedding" to better reflect what these representations currently capture. Using terms that are intuitively closer to the observed properties of the cosine similarity of text embeddings would be less likely to lead to overblown expectations.

SUPPLEMENTARY INFORMATION

The associated code is included in the supplementary archive. It contains the benchmark datasets of concept and proposition pairs with well-defined semantic entailment relationships, prompt templates used for constructing the dataset, code for training linear projections, summary reports and the code used to generate them.

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

APPENDIX

## A    DERIVATION OF ENTAILMENT SIMILARITY MEASURES

Below we will define semantic entailment similarity separately for conceptual entities (corresponding to noun phrases) and conceptual relations (i.e. predicates) and use these definitions to propose a semantic entailment measure for propositions composed of at most a subject, object and a predicate. For the sake of brevity, we will refer to conceptual entities as "concepts" and to conceptual relations as "relations".

Based on the logical framework of conceptual semantics proposed in a concurrent submission and reproduced in Appendix B, concepts $C_1$ and $C_2$ are equivalent if their semantic fields (sets of referents) are identical.

$$
\begin{aligned}
(C_i \equiv C_j) &\Leftrightarrow (R_{C_i} = R_{C_j}) \\
(R_{C_i} = R_{C_j}) &\Leftrightarrow \forall r \in R.(d(r, C_i) = d(r, C_j)))
\end{aligned}
\tag{3}
$$

where $C_1$ and $C_2$ are sets of selection criteria for the two concepts, $R_{C_i}$ is the semantic field of concept $i$ and $d(r, C_i)$ calls a decision function $d$ that returns true if the referent $r$ satisfies selection criteria $C_i$. Selection criteria are the logical equivalent to a dictionary definition of a concept since selection criteria form a set of propositions that must be satisfied by the referents of the concept.

### A.1    ENTAILMENT SIMILARITY OF CONCEPTS

Based on 3 we can derive a generic expression for the entailment similarity between concepts $C_1$ and $C_2$ based on the overlap between the semantic fields $R_{C_i}$ and $R_{C_j}$. The precise value of the semantic entailment similarity between two concepts will be available to us on the condition that we are able to exhaustively enumerate the referents of both concepts.

$$
se(C_1, C_2, \alpha, \beta) = \frac{|R_{C_1} \cap R_{C_2}|}{|R_{C_1}| + \alpha|R_{C_1} - R_{C_2}| + \beta|R_{C_2} - R_{C_1}|}
\tag{4}
$$

Setting $\alpha = \beta = 0$ will yield the special case of asymmetric entailment similarity relative to $C_1$

$$
se_{\text{asym}}(C_1, C_2) = se(C_1, C_2, 0, 0) = \frac{|R_{C_1} \cap R_{C_2}|}{|R_{C_1}|}.
\tag{5}
$$

Thus, given $R_{C_2} \subset R_{C_1}$ (i.e. concept $C_2$ is a special case of $C_1$) then $se_{\text{asym}}(C_1, C_2) < 1$ because $|R_{C_1} \cap R_{C_2}| < |R_{C_1}|$. For example, if $C_{bird}$ is the definition of the concept *bird* and $C_{\text{seabird}}$ is the definition of *seabird* and given that all seabirds are birds but not all birds are seabirds we will have $se_{\text{asym}}(C_{\text{bird}}, C_{\text{seabird}}) < 1$ and $se_{\text{asym}}(C_{\text{seabird}}, C_{\text{bird}}) = 1$. If we were to express this in terms of the implication relation we might state that $(bird \implies seabird) \sim se_{\text{asym}}(C_{\text{seabird}}, C_{\text{bird}}) = 1$ and $(seabird \implies bird) \sim se_{\text{asym}}(C_{\text{bird}}, C_{\text{seabird}}) < 1$.

An intuitive interpretation of these expressions would be that by referring to the concept *bird* we are also referring to all *seabirds* so there is a complete (full) implication from *bird* to *seabird* corresponding to value 1. On the other hand, by referring to *seabird* we are implicating only a subset of all birds hence the implication from *seabird* to *bird* is incomplete (partial) corresponding to a value $< 1$.

Setting $\alpha = 0, \beta = 1$ will yield the special case of symmetric entailment similarity that is insensitive to the order of arguments:

$$
se_{\text{sym}}(C_1, C_2) = se_{\text{sym}}(C_2, C_1) = se(C_1, C_2, 0, 1) = \frac{|R_{C_1} \cap R_{C_2}|}{|R_{C_1} \cup R_{C_2}|}.
\tag{6}
$$

Symmetric entailment similarity measure is equivalent to the fraction of the union of referents $R_{C_1} \cup R_{C_2}$ that overlaps between $R_{C_1}$ and $R_{C_2}$. It is also a stricter similarity measure than its asymmetric sibling as it requires a high degree of overlap between the referents on both sides to reach higher values.

## A.2 ENTAILMENT SIMILARITY OF PREDICATES

Predicates can be viewed as relations that accept either one or two concepts as arguments. The arguments can be named after their grammatical role as a "subject" and an "object". Predicates can be grouped into relational domains based on their logical relationships of implication and contradiction. Any **relational domain** is partitioned into subsets of predicates that are mutually contradictory between the subsets and equivalent within the subsets in the context of equivalent arguments (i.e. when applied to equivalent subjects and objects).

Relational domain $D = \{p_1, ..., p_n\}$ is a countable set of predicates $p_i : i \in \{1, ..., n\}$ where there exists a partition $P \vdash D$ such that $P = \{A_1, ..., A_m\}$ where $2 \leq m \leq 3 \leq n$ so that $A_j \subset D : j \in \{1, ..., m\}$ and $\forall i, j, i \neq j : A_i \cap A_j = \emptyset$. Importantly, $\forall p_i, p_j, i \neq j : (p_i \in A_k \wedge p_j \in A_k) \implies (p_i \iff p_j)$.

Predicate $p_i \in A_j$ is asserted in semantic context $T = \{S, O\}$ by $p_i(S, O)$ where $S$ and $O$ are the subject and the object. Given a relational domain $D$ that is partitioned into $m$ subsets of mutually exclusive predicates, assertion of $p_i(S, O)$ will contradict predicates $\bigcup_{k=1}^{m} A_k : k \neq j$ in context $T$.

Semantic entailment between predicates is given by the following piecewise-defined function:

$$se_{\text{pred}}(p_1, p_2) = \begin{cases} 1 & \text{if } \exists i \in \{1, \ldots, m\} : p_1, p_2 \in A_i \\ 0 & \text{if } p_1 \in D \wedge p_2 \notin D \\ -1 & \text{if } \exists i, j \in \{1, \ldots, m\} : p_1 \in A_i \wedge p_2 \in A_j \wedge i \neq j. \end{cases} \quad (7)$$

Thus, in a fixed context, semantic entailment between predicates is 1 if they are equivalent, 0 if they are neutral (they belong to different relational domains) and -1 if they are contradictory.

## A.3 ENTAILMENT SIMILARITY OF PROPOSITIONS

We are finally ready to propose a measure for the semantic entailment between propositions. Proposition is an assertion of a predicate within a context specified by a subject and an object. Both subject and object are optional, but one of them must be present. For example, sentences in passive voice correspond to subjectless propositions while unary predicates (e.g., *sleeps(Alice)*) do not require an object. Semantic entailment similarity between propositions $s_1$ and $s_2$ is given by

$$se(s_1, s_2) = se_{\text{asym}}(S_1, S_2) * se_{\text{asym}}(O_1, O_2) * se_{\text{pred}}(p_1, p_2) \quad (8)$$

where
$s_1 = p_1(S_1, O_1)$
$s_2 = p_2(S_2, O_2).$

In contexts where $S$ or $O$ is missing, the corresponding term will be ignored. Semantic entailment measure between propositions $s_1$ and $s_2$ exhibits the following properties:

$$\begin{aligned} &\text{equivalence:} \\ (se(s_1, s_2) = 1) &\iff (S_1 \equiv S_2 \wedge O_1 \equiv O_2 \wedge p_1 \iff p_2) \\ &\text{contradiction:} \\ (se(s_1, s_2) = -1) &\iff (S_1 \equiv S_2 \wedge O_1 \equiv O_2 \wedge (p_1 \wedge p_2 \implies \bot)) \\ &\text{neutrality:} \\ (se(s_1, s_2) = 0) &\iff (R_{S_1} \cup R_{S_2} = \emptyset) \vee (R_{O_1} \cup R_{O_2} = \emptyset) \vee (p_1 \| p_2) \\ &\text{partial equivalence:} \\ 0 < se(s_1, s_2) < 1 &\iff W \wedge (p_1 \iff p_2) \\ &\text{partial contradiction:} \\ -1 < se(s_1, s_2) < 0 &\iff W \wedge (p_1 \wedge p_2 \implies \bot) \end{aligned} \quad (9)$$

where $R_C$ is the semantic field of concept $C$ and $p_1 \| p_2 \iff (p_1 \in D \wedge p_2 \notin D)$ and the condition for partial overlap of referents is $W = (R_{S_1} \subset R_{S_2} \vee R_{O_1} \subset R_{O_2}) \wedge (R_{S_1} \cap R_{S_2} \neq \emptyset) \wedge (R_{O_1} \cap R_{O_2} \neq \emptyset)$.

# B  FORMAL FRAMEWORK OF CONCEPTUAL SEMANTICS (REPRODUCED FROM A CONCURRENT SUBMISSION)

## B.1  CONCEPT

Concept $C$ is a set of selection criteria $C = \{c_1, c_2, ..., c_n\}$ for its referents $R_C$.

Let selection criterion $c_i$ be a proposition corresponding to a linguistic statement[1] about some entity $x$. Given an entity $r$ and selection criteria $C$, a decision function $d(r, C) \rightarrow \{true, false\}$ will output $true$ only if all selection criteria $c_i \in C$ apply to $r$. Thus, the application of $d$ to its arguments corresponds to a logical conjunction of the selection criteria $C$ when evaluated in the context of $r$:

$$d(r, C) \equiv \left( r \vdash \bigwedge_{c_i \in C} \right). \tag{10}$$

If $r$ satisfies the selection criteria $C$ then $r$ is considered a referent of $C$.

### B.1.1  SEMANTIC FIELD

The set of referents $R_C$ of concept $C$ is a collection of imagined entities $R_C = \{r_1, r_2, ..., r_n\}$ that satisfy selection criteria C.

$$C \implies R_C : \forall r_i \in R_C.(d(r_i, C) = \text{true}) \tag{11}$$

In subsequent text we will refer to $R_C$ as the **semantic field** of concept $C$.

## B.2  IDENTITY OF SELECTION CRITERIA

Selection criteria $c_i$ and $c_j$ are identical if and only if their symbolic representations (as linguistic terms) are identical.

$$(c_i = c_j) \Leftrightarrow (E(\{c_i\}) = E(\{c_j\})), \tag{12}$$

where function $E$ maps a set of selection criteria to a corresponding linguistic form in a suitable language (e.g. English).

## B.3  EQUIVALENCE OF SELECTION CRITERIA

Selection criteria $c_i$ and $c_j$ are equivalent if and only if they imply the same set of referents

$$(c_i \equiv c_j) \Leftrightarrow (c_i \implies R_{c_i}) \wedge (c_j \implies R_{c_j}) \wedge (R_{c_i} = R_{c_j}). \tag{13}$$

One might also want to distinguish between the logical and practical aspects of the equivalence of selection criteria. Thus, two sets of selection criteria are logically equivalent if they imply each other. For example, given the formulations $C_1$ and $C_2$ of the concept *human protein coding gene* as:

$$C_1 = \{c_1 : \text{"it is a human gene"}, c_2 : \text{"it encodes protein"}\}$$
$$C_2 = \{c_3 : \text{"it is a gene"}, c_4 : \text{"it is found in human"}, c_5 : \text{"it encodes protein"}\}.$$

One can safely state that $C_1 \Leftrightarrow C_2$ because $c_2 = c_5$ and $c_1 \Leftrightarrow c_3 \wedge c_4$.

---

[1]Linguistic statements (e.g., "it will rain tomorrow") are the subset of linguistic expressions that can be true or false. Linguistic expressions (e.g., "white cat") are words and grammatically valid word combinations that are a subset of finite combinations of symbols from an alphabet.

It follows that two logically equivalent formulations even though not identical refer to the same set of referents in all possible contexts. On the other hand, it is also possible that two definitions that are not logically equivalent refer to the same referents in some context $R' \subset R$ but not necessarily in all possible subsets of $R$.

### B.4 LINGUISTIC INTERPRETATION

Linguistic interpretation is a chain of reference where linguistic term $t$ refers to concept $C$ that implies referents $R_C$.

$$t \implies C \implies R_C. \tag{14}$$

Thus, expression equation 14 establishes a sequence where a symbol refers to a concept that refers to its semantic field. The implication from symbol to concept is based on arbitrary association as evidenced by translatability between natural languages while the implication between a concept and its referents is determined by the selection criteria.

### B.5 CONCEPTUAL IDENTITY

Concepts $C_i$ and $C_j$ are identical if and only if they can be decomposed into equivalent selection criteria so that there exists a bijection between the sets of selection criteria $C_i$ and $C_j$ where $\{c_k, c_l\}$ : $c_k \in C_i, c_l \in C_j$ imply the same set of referents.

$$(C_i = C_j) \Leftrightarrow \begin{matrix} f : C_i \to C_j, f(g(c_l)) = c_l \\ g : C_j \to C_i, g(f(c_k)) = c_k, \end{matrix} \tag{15}$$

where $c_k \in C_i, c_l \in C_j, \forall r \in R.(d(r, \{c_k\}) = d(r, \{c_l\}))$.

### B.6 CONCEPTUAL EQUIVALENCE

Concepts $C_1$ and $C_2$ are equivalent if their semantic fields are identical.

$$\begin{aligned} (C_i \equiv C_j) &\Leftrightarrow (R_{C_i} = R_{C_j}) \\ (R_{C_i} = R_{C_j}) &\Leftrightarrow \forall r \in R.(d(r, C_i) = d(r, C_j))(\text{by } 11) \end{aligned} \tag{16}$$

We can imagine a concept $C_1 = \{c_1\}$ with one selection criterion which yields a set of referents $R_{C_1}$ and an equivalent concept $C_2 = \{c_2, c_3\}$ that yields an identical set of referents $R_{C_2} = R_{C_1}$.

### B.7 PROPERTIES OF THE CONCEPTUAL SEMANTICS FRAMEWORK

Proofs of the properties outlined below are provided as Lean 4 code in the supplementary archive (folder "external_resources/proofs"). The assistance of GitHub Copilot was used to suggest tactics for the proofs.

#### B.7.1 COMPOSITIONALITY OF CONCEPTS AND NOUN PHRASES

Compositionality of concepts enables addition and removal of selection criteria to specify and relax the concept by restricting and expanding its semantic field, correspondingly.

$$C' = C \cup \{c\} \implies R_{C'} \subseteq R_C(\text{by } 10 \text{ and } 11) \tag{17}$$

$$C'' = C \setminus \{c\} \implies R_C \subseteq R_{C''}(\text{by } 10 \text{ and } 11) \tag{18}$$

This additive nature of selection criteria supports conceptual hierarchies. For example, assertions "all surgeons are doctors" and "all doctors are not surgeons" imply that the referents of *surgeon* satisfy the selection criteria in *doctor*, while the referents of *doctor* do not satisfy all selection

criteria in *surgeon*. This implies that the selection criteria in *doctor* are a subset of selection criteria in *surgeon* creating a nested hierarchy between the concepts *doctor* and *surgeon*:

$$C_1 \subset C_2 \implies R_{C_2} \subset R_{C_1},$$

where $C_1$ and $C_2$ are the selection criteria of the concepts *doctor* and *surgeon*, correspondingly. The referents of *doctor* are a superset of the referents of *surgeon*.

We can use function $E$ to map from the semantic form of concept to a corresponding linguistic form e.g., in English:

$$E(C_1) = \text{"doctor"}$$
$$E(C_2) = \text{"surgeon"}$$

Productivity of language (the ability to combine linguistic terms into distinctly meaningful expressions) is another manifestation of the compositionality of concepts. Thus, the semantic field of the expression "a white rabbit with a black hat and a pocket watch" is an intersection of the semantic fields of the terms "white", "rabbit", "with a black hat", "with a pocket watch" when interpreted according to English grammar in the given context.

$$E(\{c_1, c_2, c_3\}) = \text{"a white rabbit with a black hat and a pocket watch"}$$

where

$$E(\{c_1\}) = \text{"it is a white rabbit"}$$
$$E(\{c_2\}) = \text{"it has a black hat"}$$
$$E(\{c_3\}) = \text{"it has a pocket watch"}$$

### B.7.2 COMPOSITIONALITY OF MEANING

Contradictory selection criteria result in a concept with an empty semantic field. Such concepts can be regarded as nonsensical or meaningless (as opposed to meaningful), because there is no context where they can have referents.

The size of semantic field $R_C$ of concept $C$ is null when there exists in $C$ at least one pair of selection criteria $c_i, c_j$ such that there are no referents $r$ that satisfy $c_i$ and $c_j$ simultaneously:

$$\exists c_i \exists c_j \in C : \forall r \in R, \neg(d(r, \{c_i\}) \wedge d(r, \{c_j\}))) \implies (R_C = \emptyset) \text{(by 10 and 11)} \tag{19}$$

### B.7.3 COMPOSITIONALITY OF LOGICAL OPERATIONS IN NATURAL LANGUAGE

It is plausible that semantic fields of size null as exemplified above have also practical implications. Concepts can be combined with logical operators in natural language to expand or restrict the semantic field of a phrase and it seems that the interpretation of the result is contingent on the anticipated semantic field size.

For example, combination of adjectives with "or" in a noun phrase implies the union of their semantic fields:

$$C_i \vee C_j \implies R_{C_i} \cup R_{C_j} \tag{20}$$

Example 1: the set of referents of "sweet or sour sauce" coincides with the union of the referents of "sweet sauce" and "sour sauce".

Combination of non-contradictory adjectives with "and" implies an intersection of their semantic fields:

$$C_i \wedge C_j \implies R_{C_i} \cap R_{C_j} \tag{21}$$

Example 2: the set of referents of "sweet and sour sauce" is the intersection of the sets of referents of "sweet sauce" and "sour sauce".

However, a conjunction of contradictory adjectives in a noun phrase is typically interpreted as a union operation.

Example 3: the set of referents of "hot and cold beverages" is the union of the sets of referents of "hot beverages" and "cold beverages".

Likewise, the semantic field of a conjunction of nouns (e.g. "diamonds and pearls", "Alice and Bob") is to be interpreted as the union of the arguments since the intersection (e.g. the set of referents satisfying both the selection criteria of *diamonds* and *pearls*) is anticipated to be null.

## C  DATASET EXAMPLES

### C.1  RELATIONAL DOMAINS

Below are three examples of relational domain definitions. Each relational domain contains two or three subsets of predicates that are mutually contradictory between the subsets and synonymous within the subsets.

```
# Relational domain 1
[
    {
        "title": "decrease",
        "rationale": "All predicates that imply a decrease in a \
        quantified measure.",
        "predicates": [
            "attenuated",
            "dampened",
            "deactivated",
            "inactivated",
            "decreased",
            "diminished",
            "inhibited",
            "lowered",
            "reduced",
            "repressed",
            "suppressed",
            "silenced",
            "was downregulated",
            "was reduced",
            "downregulated"
        ]
    },
    {
        "title": "increase",
        "rationale": "All predicates that imply a increase in a quantified \
        measure.",
        "predicates": [
            "activated",
            "amplified",
            "augmented",
            "elevated",
            "increased",
            "raised",
            "promoted",
            "stimulated",
            "boosted",
            "upregulated",
            "was upregulated",
            "was increased",
```

```
                        "was elevated"
                    ]
            },
            {
                "title": "no_increase_or_decrease",
                "rationale": "All predicates that imply a lack of quantifiable effect \
                on a measure.",
                "predicates": [
                    "did not alter"
                ]
            }
        ]

        # Relational domain 2
        [
            {
                "title": "broadened",
                "rationale": "All predicates that imply an increase or expansion in \
                scope, width, or range.",
                "predicates": [
                    "broadened",
                    "expanded"
                ]
            },
            {
                "title": "narrowed",
                "rationale": "All predicates that imply a decrease, restriction, or \
                narrowing in scope, width, or range.",
                "predicates": [
                    "narrowed",
                    "restricted",
                    "confined",
                    "limited",
                    "curbed",
                    "restrained"
                ]
            }
        ]

        # Relational domain 3
        [
            {
                "title": "increase_in_speed",
                "rationale": "All predicates that imply an increase or acceleration in \
                speed or rate.",
                "predicates": [
                    "accelerated",
                    "sped up"
                ]
            },
            {
                "title": "decrease_in_speed",
                "rationale": "All predicates that imply a decrease, deceleration, or \
                delay in speed or rate.",
                "predicates": [
                    "slowed down",
                    "decelerated",
                    "delayed"
                ]
```

```
      }
   ]
```

## C.2 ANNOTATIONS OF CONCEPTUAL ENTITIES AND RELATIONS

Below are three examples of dataset entries with two concept labels ("a" and "b") with corresponding annotations of semantic entailment ("a - b" and "b - a"), symbolic similarity on character and token level ("char(a  b)" and "token(a  b)") and the domain of knowledge.

```
   {
        "a": "peptide hormone",
        "b": "Gonadotropins",
        "a -> b": 1.0,
        "b -> a": 0.096,
        "char(a ~ b)": 0.5255189255189255,
        "token(a ~ b)": 0.0,
        "domain": "biology"
   }

   {
        "a": "Glycine",
        "b": "Gly",
        "a -> b": 1.0,
        "b -> a": 1.0,
        "char(a ~ b)": 0.8666666666666668,
        "token(a ~ b)": 0.5833333333333333,
        "domain": "chemistry"
   }

   {
        "a": "Interneurons",
        "b": "Amacrine Cells",
        "a -> b": 1.0,
        "b -> a": 0.2,
        "char(a ~ b)": 0.5246031746031746,
        "token(a ~ b)": 0.0,
        "domain": "medicine"
   }
```

Below are three examples of dataset entries with two predicates labels ("a" and "b") with corresponding annotations of semantic entailment ("a - b" and "b - a"), symbolic similarity and relational domain.

```
   {
        "a": "inhibited",
        "b": "silenced",
        "a -> b": 1.0,
        "b -> a": 1.0,
        "char(a ~ b)": 0.6481481481481481,
        "token(a ~ b)": 0.0,
        "domain": "decrease"
   }

   {
        "a": "was capable",
        "b": "was unable",
        "a -> b": -1.0,
        "b -> a": -1.0,
        "char(a ~ b)": 0.9054545454545454,
```

```
1080        "token(a ~ b)": 0.41666666666666663,
1081        "domain": "has_capacity/lacks_capacity"
1082    }
1083
1084    {
1085        "a": "did not contain",
1086        "b": "was capable",
1087        "a -> b": 0.0,
1088        "b -> a": 0.0,
1089        "char(a ~ b)": 0.4909090909090909,
1090        "token(a ~ b)": 0.0,
1091        "domain": "not_contain/has_capacity"
1092    }
```

## C.3  PROPOSITIONS DATASET

Below are three examples of dataset entries for proposition pairs with well-defined semantic entailment relationships. Domain refers to the proposition construction schema from Appendix E.

```
    {
        "a": "[Glucagon-like peptide-2] [displayed] [Sodium-Potassium-Chloride \
        Cotransporter] activity in the intestinal epithelium.",
        "b": "[GLP-2] [displayed] [Sodium-Potassium-Chloride Cotransporter] \
        activity in the intestinal epithelium.",
        "a -> b": 1.0,
        "b -> a": 1.0,
        "char(a ~ b)": 0.8324621238325839,
        "token(a ~ b)": 0.8400932400932402,
        "domain": "1.1"
    }

    {
        "a": "[Piperidine] [displayed] binding to [Glycine receptor] in the \
        hippocampal neurons.",
        "b": "[hexahydroazepine] [did not display] binding to [Glycine receptor] \
        in the hippocampal neurons.",
        "a -> b": -1.0,
        "b -> a": -1.0,
        "char(a ~ b)": 0.8092520757654385,
        "token(a ~ b)": 0.6251572327044025,
        "domain": "3.1"
    }

    {
        "a": "[Malate dehydrogenase] [served as] a cofactor for [Nitrogen trioxide]\
        in the reaction mixture.",
        "b": "[Malate dehydrogenase] [served as] a cofactor for [Werner syndrome \
        RecQ like helicase] in the reaction mixture.",
        "a -> b": 0.0,
        "b -> a": 0.0,
        "char(a ~ b)": 0.9050815305400779,
        "token(a ~ b)": 0.7116244411326378,
        "domain": "4.2"
    }
```

## D PROMPT TEMPLATES

This appendix contains the prompt templates used for data collection and processing in our study.

### D.1 VALIDATE CONCEPT LABELS

You are given two different labels for the same concept. Make sure that these labels are both common enough and, in practice, unambiguously associated with the concept across various contexts. Do not entertain highly theoretical possibilities as a reason for suggesting an alternative label. Please suggest better alternatives for distinct labels. Format the response as valid JSON (do not add markdown formatting). Return an empty object {} if the labels are already well disambiguated.

**Example:**

Input:

```
{
    "domain": "chemistry",
    "label1": "Vanillin",
    "label2": "C8H8O3",
    "jaro_winkler_similarity": 0.0
}
```

Output:

```
{
    "domain": "chemistry",
    "label1": "Vanillin",
    "label2": "4-hydroxy-3-methoxybenzaldehyde",
    "jaro_winkler_similarity": 0.0,
    "reason": "C8H8O3 is an ambiguous label as it can also refer to \
    salicylic acid."
}
```

### D.2 EXTRACT PREDICATES FROM ABSTRACT

Extract predicates (verbs with associated modifiers) related to the findings of the study from the following biomedical research abstracts into a json form. Return only JSON and nothing else. Do not use pronouns in context (e.g., replace "some of these markers" with the markers mentioned in the abstract). Leave the subject empty for sentences in passive voice. Do not add markdown formatting to the output. Make sure to escape any double quotes (e.g., change " to \") in the abstract contents.

**Example:**

Abstract: Although we have previously shown that I-IFN, activated by interferon regulatory factor 3 (IRF3), plays an essential role in limiting prion invasion, the precise mechanisms underlying its protective effects remain unclear. Here, using in vivo and ex vivo prion infection models, we discovered that 2'-5' oligoadenylate synthetase 1a (Oas1a), an interferon-stimulated gene downstream of the I-IFN receptor, inhibits prion invasion at an early stage...

JSON:

```
[
  {
    "subject": "gene/protein",
    "predicate": "inhibits",
    "object": "process",
    "context": {
      "gene/protein": "2'-5' oligoadenylate synthetase 1a (Oas1a)",
      "process": "prion invasion at an early stage"
    }
```

```
     },
     {
       "subject": "loss of gene/protein",
       "predicate": "significantly accelerates",
       "object": "process",
       "context": {
         "gene/protein": "Oas1a",
         "process": "prion disease progression"
       }
     }
   ]
```

### D.3  IDENTIFY RELATIONAL DOMAINS

Relational domain is a partitioning of a set of predicates into mutually contradictory subsets. Predicates within each subset of the partition imply each other while predicates in different subsets are contradictory. When evaluated in the context of a fixed semantic subject and object, contradictory predicates will yield contradictory propositions. Predicates that belong to different relational domains yield independent propositions when applied to a fixed subject and object.

You are a biomedical research, semantic analysis and predicate logic expert. Your task is to partition a given list of predicates into relational domains. Predicates that do not imply or contradict each other belong to different relational domains. Your task is to identify unique relational domains based on the given list of predicates. Add predicates to the relational domain in the third person singular form. Do not create a new relational domain if the predicate fits into an existing one. Return the identified relational domains as JSON and nothing else. Do not add markdown formatting to the output. Do not include anthropomorphic predicates such as "show", "demonstrated", "suggested", "indicates", "revealed", "identified" etc. DO NOT REPEAT relational domains, make sure each predicate is found in only one relational domain.

### D.4  GENERATE PROPOSITIONS

You are a biomedical research expert highly skilled in the semantic analysis of propositions. You will be given a random selection of subjects, objects and predicates and a task to form two contextually plausible sentences with them. For each sentence you need to pick one subject, one object and a predicate while following the constraints given in the task instruction. Predicates are be given as relational domains that are lists composed of several subsets of contradictory predicates. You have to ignore predicates without synonyms if asked to provide sentences with synonymous but distinct predicates. The generated sentences DO NOT have to be factually correct, just contextually plausible. In case you introduce any modifying phrases to the subject or object, make sure the phrase is represented in both sentences to ensure strict adherence to the task instructions. In the generated sentences, make sure to enclose in square brackets each subject, object and predicate (verb) that was used from the available options. If the available subjects, objects and predicates do not seem to yield contextually plausible sentences then return an empty JSON object to request another set of concepts. Return a valid JSON object with the two sentences and corresponding annotations. Do not include markdown formatting.

### D.5  SEMANTIC AGENT

You are a coding agent with access to a limited set of language features following Python syntax. You will be given a task which you need to solve using the given language features. You can store variables containing any information required for solving a task. The `input` variable is read only and it contains source information for the task. Make sure to output only valid Python code (you can include

reasoning traces as comments accompanying the code). In general, try to reuse the code examples below as much as possible and write as little extra business logic as possible. Your main goal is to reason about predicate semantics and generate simple and concise code instructions.

Available language features:

1. any operations with variables (including adding and modifying variables)
2. `find` function with usage `find(query:str, k:int=3)` that will return k most similar matches to the query from the `input` variable.
3. `observe` function with usage `observe(variable:Any)` that will return up to k elements of the variable (where each element can be at most 500 characters long).

Background: Elements in each sublist contain related predicates that form a relational domain. Relational domain is a partitioning of predicates into subsets so that any two predicates from different subsets are always contradictory while any two predicates from the same subset always imply each other. Unrelated (mutually independent) predicates belong to different relational domains. Each subset in a relational domain is characterized by a rationale that outlines the common theme for the listed predicates.

Task: Your task is to merge all sublists (relational domains) in `input` that have similar rationales while making sure that the predicates within each sublists are synonymous in the context of the sublist's rationale.

## D.6 RELATIONAL DOMAIN CONSISTENCY

You will be given a list of relational domains composed of predicates and your task is to point out any inconsistencies in or between the relational domains. Each relational domain consists of one to three subsets of predicates that are mutually contradictory between the subsets and mutually implicative within each subset. Different relational domains should be independent of each other so that when we apply two predicates from different relational domains in the same context (e.g. to the same subject and object) we will always get propositions that do not imply or contradict each other in the given context.

```
<relational_domains>
{relational_domains}
</relational_domains>
```

Your task: 1. Highlight any predicates that are inconsistent with their companions within a subset of a relational domain. 2. Highlight any predicate subsets of a single relational domain that are not mutually contradictory. 3. Highlight any relational domains that are not mutually independent.

For each issue, provide argumentation with corresponding examples.

## E PROPOSITION-LEVEL CATEGORIES OF SEMANTIC ENTAILMENT

### E.1 EQUIVALENT PROPOSITIONS (TWO-WAY IMPLICATION)

*Propositions with identical or synonymous subjects, objects and predicates*

1.1 Synonymous but distinct subjects, identical objects and predicates
- A: [Chlorine dioxide] [increased] [rhamnose] levels in bacteria
- B: [ClO2] [increased] [rhamnose] levels in bacteria

1.2 Synonymous objects, identical subjects and predicates
- A: [Chlorine dioxide] [suppressed] [Bromine monoxide] levels in bacteria
- B: [Chlorine dioxide] [suppressed] [BrO] levels in bacteria

1.3 Synonymous predicates, identical subjects and objects

- A: [ClO2] [increased] [rhamnose] levels in bacteria
- B: [ClO2] [raised] [rhamnose] levels in bacteria

1.4 Synonymous subjects and objects, identical predicates

- A: [ClO2] [increased] [rhamnose] levels in bacteria
- B: [Chlorine dioxide] [increased] [Rha] levels in bacteria

1.5 Synonymous subjects and predicates, identical objects

- A: [ClO2] [raised] [Rha] levels in bacteria
- B: [Chlorine dioxide] [increased] [Rha] levels in bacteria

1.6 Synonymous objects and predicates, identical subjects

- A: [ClO2] [increased] [Rha] levels in bacteria
- B: [ClO2] [raised] [rhamnose] levels in bacteria

1.7 Synonymous subjects, objects and predicates

- A: [ClO2] [increased] [Rha] levels in bacteria
- B: [Chlorine dioxide] [raised] [rhamnose] levels in bacteria

## E.2 ONE-WAY IMPLICATION

*Propositions that logically entail each other in one direction because one subject, object or predicate is a subset of the other*

2.1 One-way implication between subjects, identical objects and predicates

- A: [All human cells] [express] [XBP1]
- B: [B-Lymphocytes] [express] [XBP1]

2.2 One-way implication between objects, identical subjects and predicates

- A: [Phosphoenolpyruvate] level [is suppressed] by [artemisinin derivatives] in mice
- B: [Phosphoenolpyruvate] level [is suppressed] by [artenimol] in mice

2.3 One-way implication between predicates, identical subjects and objects

- A: [Phosphoenolpyruvate] level [is affected] by [artemisinin derivatives] in mice
- B: [Phosphoenolpyruvate] level [is suppressed] by [artemisinin derivatives] in mice

## E.3 CONTRADICTORY PROPOSITIONS

*Propositions with contradictory predicates and equivalent subjects or objects*

3.1 Synonymous subjects, contradictory predicates and identical objects

- A: [Chlorine dioxide] [increased] [rhamnose] levels in bacteria
- B: [ClO2] [decreased] [rhamnose] levels in bacteria

3.2 Synonymous objects, contradictory predicates and identical subjects

- A: [Phosphoenolpyruvate] level [is affected] by [rhamnose] levels in bacteria
- B: [Phosphoenolpyruvate] level [is not impacted] by [Rha] levels in bacteria

3.3 Synonymous subjects and objects, contradictory predicates

- A: [ClO2 dioxide] [increased] [rhamnose] levels in bacteria
- B: [Chlorine dioxide] [decreased] [Rha] levels in bacteria

## E.4 UNRELATED PROPOSITIONS

*Propositions with either unrelated subjects, objects, or predicates*

4.1 Unrelated subjects, identical objects and predicates

- A: [Chlorine dioxide] [increased] [rhamnose] levels in bacteria

- B: [Nitrogen dioxide] [increased] [rhamnose] levels in bacteria

### 4.2 Unrelated objects, identical subjects and predicates

- A: [Chlorine dioxide] [increased] [glucose] levels in bacteria
- B: [Chlorine dioxide] [increased] [rhamnose] levels in bacteria

### 4.3 Unrelated predicates, identical subjects and objects

- A: [NSAIDs] [increase] the expression of [Zinc Transporters]
- B: [NSAIDs] [bind] to [Zinc Transporters]

# F MODEL RANKINGS ON SEMANTIC ENTAILMENT AND SYMBOLIC SIMILARITY ESTIMATION FROM THREE TRAINING RUNS OF LINEAR PROJECTIONS

Table 4: Summary of rankings based on semantic entailment similarity estimation accuracy (run 1)

| Model | Sum of Ranks | Mean Rank | Times in Top Third | p | q |
|---|---|---|---|---|---|
| **text-embedding-3-large** | 69 | 2.46 | 27 | 0 | **0** |
| **e5-mistral-7b-instruct** | 77 | 2.75 | 26 | $1 \times 10^{-10}$ | $\mathbf{2.49 \times 10^{-9}}$ |
| **gte-Qwen2-1.5B-instruct** | 123 | 4.39 | 20 | $4.26 \times 10^{-5}$ | **0.001** |
| **Qwen3-Embedding-8B** | 128 | 4.57 | 18 | 0.001 | **0.010** |
| gte-Qwen2-7B-instruct | 179 | 6.39 | 14 | 0.050 | 0.501 |
| bilingual-embedding-large | 195 | 6.96 | 7 | 0.874 | 1 |
| cohere.embed-multilingual-v3 | 202 | 7.21 | 9 | 0.623 | 1 |
| stella_en_1.5B_v5 | 203 | 7.25 | 6 | 0.943 | 1 |
| text-embedding-3-small | 225 | 8.04 | 7 | 0.874 | 1 |
| bge-m3 | 252 | 9.00 | 6 | 0.943 | 1 |
| GIST-large-Embedding-v0 | 298 | 10.6 | 0 | 1 | 1 |
| Qwen3-Embedding-0.6B | 321 | 11.5 | 0 | 1 | 1 |
| snowflake-arctic-embed-l-v2.0 | 350 | 12.5 | 0 | 1 | 1 |
| gte-base-en-v1.5 | 364 | 13.0 | 0 | 1 | 1 |
| gte-large-en-v1.5 | 374 | 13.4 | 0 | 1 | 1 |

Table 5: Summary of rankings based on semantic entailment similarity estimation accuracy (run 2)

| Model | Sum of Ranks | Mean Rank | Times in Top Third | p | q |
|---|---|---|---|---|---|
| **e5-mistral-7b-instruct** | 72 | 2.57 | 26 | $1 \times 10^{-10}$ | $\mathbf{4.98 \times 10^{-9}}$ |
| **text-embedding-3-large** | 83 | 2.96 | 25 | $1.2 \times 10^{-9}$ | $\mathbf{2.99 \times 10^{-8}}$ |
| **gte-Qwen2-1.5B-instruct** | 131 | 4.68 | 19 | $1.97 \times 10^{-4}$ | **0.003** |
| Qwen3-Embedding-8B | 152 | 5.43 | 16 | 0.008 | 0.101 |
| gte-Qwen2-7B-instruct | 176 | 6.29 | 12 | 0.191 | 1 |
| stella_en_1.5B_v5 | 194 | 6.93 | 8 | 0.765 | 1 |
| bilingual-embedding-large | 195 | 6.96 | 8 | 0.765 | 1 |
| cohere.embed-multilingual-v3 | 207 | 7.39 | 10 | 0.464 | 1 |
| bge-m3 | 211 | 7.54 | 9 | 0.623 | 1 |
| text-embedding-3-small | 212 | 7.57 | 6 | 0.943 | 1 |
| GIST-large-Embedding-v0 | 317 | 11.3 | 1 | 1.000 | 1 |
| Qwen3-Embedding-0.6B | 333 | 11.9 | 0 | 1 | 1 |
| gte-base-en-v1.5 | 354 | 12.6 | 0 | 1 | 1 |
| snowflake-arctic-embed-l-v2.0 | 357 | 12.8 | 0 | 1 | 1 |
| gte-large-en-v1.5 | 366 | 13.1 | 0 | 1 | 1 |

Table 6: Summary of rankings based on semantic entailment similarity estimation accuracy (run 3)
Same contents as in Table 2

Table 7: Summary of rankings based on symbolic similarity estimation accuracy (run 1)

| Model | Sum of Ranks | Mean Rank | Times in Top Third | p | q |
|---|---|---|---|---|---|
| **bge-m3** | 83 | 2.96 | 24 | $1.55 \times 10^{-8}$ | $\mathbf{7.71 \times 10^{-7}}$ |
| **snowflake-arctic-embed-l-v2.0** | 132 | 4.71 | 18 | 0.000785 | 0.020 |
| cohere.embed-multilingual-v3 | 167 | 5.96 | 15 | 0.022 | 0.268 |
| stella_en_1.5B_v5 | 170 | 6.07 | 15 | 0.022 | 0.268 |
| text-embedding-3-small | 178 | 6.36 | 14 | 0.050 | 0.501 |
| Qwen3-Embedding-0.6B | 192 | 6.86 | 11 | 0.314 | 1 |
| bilingual-embedding-large | 195 | 6.96 | 8 | 0.765 | 1 |
| gte-Qwen2-1.5B-instruct | 197 | 7.04 | 12 | 0.191 | 1 |
| gte-Qwen2-7B-instruct | 217 | 7.75 | 7 | 0.874 | 1 |
| text-embedding-3-large | 245 | 8.75 | 6 | 0.943 | 1 |
| GIST-large-Embedding-v0 | 284 | 10.1 | 0 | 1 | 1 |
| gte-base-en-v1.5 | 284 | 10.1 | 3 | 0.999 | 1 |
| Qwen3-Embedding-8B | 309 | 11.0 | 6 | 0.943 | 1 |
| gte-large-en-v1.5 | 325 | 11.6 | 1 | 1.000 | 1 |
| e5-mistral-7b-instruct | 382 | 13.6 | 0 | 1 | 1 |

Table 8: Summary of rankings based on symbolic similarity estimation accuracy (run 2)

| Model | Sum of Ranks | Mean Rank | Times in Top Third | p | q |
|---|---|---|---|---|---|
| **bge-m3** | 79 | 2.82 | 23 | $1.53 \times 10^{-7}$ | $\mathbf{7.62 \times 10^{-6}}$ |
| **snowflake-arctic-embed-l-v2.0** | 106 | 3.79 | 22 | $1.21 \times 10^{-6}$ | $\mathbf{3 \times 10^{-5}}$ |
| text-embedding-3-small | 179 | 6.39 | 15 | 0.022 | 0.358 |
| Qwen3-Embedding-0.6B | 182 | 6.50 | 10 | 0.464 | 1 |
| stella_en_1.5B_v5 | 184 | 6.57 | 14 | 0.050 | 0.626 |
| gte-Qwen2-1.5B-instruct | 205 | 7.32 | 8 | 0.765 | 1 |
| cohere.embed-multilingual-v3 | 206 | 7.36 | 13 | 0.104 | 1 |
| bilingual-embedding-large | 208 | 7.43 | 9 | 0.623 | 1 |
| text-embedding-3-large | 232 | 8.29 | 8 | 0.765 | 1 |
| gte-Qwen2-7B-instruct | 235 | 8.39 | 6 | 0.943 | 1 |
| gte-base-en-v1.5 | 262 | 9.36 | 5 | 0.979 | 1 |
| GIST-large-Embedding-v0 | 292 | 10.4 | 2 | 1 | 1 |
| gte-large-en-v1.5 | 294 | 10.5 | 1 | 1 | 1 |
| Qwen3-Embedding-8B | 307 | 11.0 | 4 | 0.994 | 1 |
| e5-mistral-7b-instruct | 389 | 13.9 | 0 | 1 | 1 |

Table 9: Summary of rankings based on symbolic similarity estimation accuracy (run 3)

| Model | Sum of Ranks | Mean Rank | Times in Top Third | p | q |
|---|---|---|---|---|---|
| **bge-m3** | 75 | 2.68 | 26 | $1 \times 10^{-10}$ | $\mathbf{4.98 \times 10^{-9}}$ |
| **snowflake-arctic-embed-l-v2.0** | 105 | 3.75 | 21 | $7.83 \times 10^{-6}$ | $\mathbf{1.95 \times 10^{-4}}$ |
| text-embedding-3-small | 165 | 5.89 | 12 | 0.191 | 1 |
| cohere.embed-multilingual-v3 | 166 | 5.93 | 14 | 0.050 | 0.834 |
| bilingual-embedding-large | 198 | 7.07 | 10 | 0.464 | 1 |
| stella_en_1.5B_v5 | 203 | 7.25 | 8 | 0.765 | 1 |
| gte-Qwen2-1.5B-instruct | 207 | 7.39 | 9 | 0.623 | 1 |
| Qwen3-Embedding-0.6B | 212 | 7.57 | 11 | 0.314 | 1 |
| text-embedding-3-large | 238 | 8.50 | 7 | 0.874 | 1 |
| gte-Qwen2-7B-instruct | 248 | 8.86 | 7 | 0.874 | 1 |
| gte-base-en-v1.5 | 257 | 9.18 | 5 | 0.979 | 1 |
| Qwen3-Embedding-8B | 296 | 10.6 | 7 | 0.874 | 1 |
| GIST-large-Embedding-v0 | 297 | 10.6 | 1 | 1 | 1 |
| gte-large-en-v1.5 | 297 | 10.6 | 2 | 1 | 1 |
| e5-mistral-7b-instruct | 396 | 14.1 | 0 | 1 | 1 |

Table 10: Model pairs used for differential entailment estimation error analysis. The model on the left exhibited better performance in entailment estimation.

| Comparison type | Better (at entailment) | Worse |
|---|---|---|
| Siblings (size) | text-embedding-3-large | text-embedding-3-small |
| Siblings (size) | Qwen3-Embedding-8B | Qwen3-Embedding-0.6B |
| Siblings (size) | gte-Qwen2-1.5B-instruct | gte-Qwen2-7B-instruct |
| Semantic vs symbolic | text-embedding-3-large | snowflake-arctic-embed-l-v2.0 |
| Semantic vs symbolic | e5-mistral-7b-instruct | snowflake-arctic-embed-l-v2.0 |
| Semantic vs symbolic | gte-Qwen2-1.5B-instruct | snowflake-arctic-embed-l-v2.0 |
| Semantic vs symbolic | text-embedding-3-large | bge-m3 |
| Semantic vs symbolic | e5-mistral-7b-instruct | bge-m3 |
| Semantic vs symbolic | gte-Qwen2-1.5B-instruct | bge-m3 |

Table 11: Proposition entailment categories ranked by descending estimation error differential

| Category | Times in Top Third | p | q | Mean estimate (model 1) | Mean estimate (model 2) | Mean error (model 1) | Mean error (model 2) |
|---|---|---|---|---|---|---|---|
| **1.2** | 53 | 0 | **0** | 0.912 | 0.858 | 0.088 | 0.142 |
| **3.1** | 46 | $1 \times 10^{-10}$ | $\mathbf{2.49 \times 10^{-9}}$ | -0.504 | -0.188 | 0.496 | 0.812 |
| **1.7** | 36 | $8.87 \times 10^{-5}$ | **0.001** | 0.797 | 0.701 | 0.203 | 0.299 |
| **3.2** | 32 | 0.003 | **0.039** | -0.485 | -0.211 | 0.515 | 0.789 |
| 1.4 | 31 | 0.007 | 0.065 | 0.814 | 0.728 | 0.186 | 0.272 |
| 3.3 | 28 | 0.043 | 0.359 | -0.563 | -0.364 | 0.437 | 0.636 |
| 1.1 | 25 | 0.174 | 1 | 0.881 | 0.830 | 0.119 | 0.170 |
| 1.6 | 22 | 0.441 | 1 | 0.893 | 0.847 | 0.107 | 0.153 |
| 1.5 | 9 | 1 | 1 | 0.866 | 0.825 | 0.134 | 0.175 |
| 4.3 | 7 | 1 | 1 | 0.613 | 0.715 | 0.639 | 0.725 |
| 4.1 | 6 | 1 | 1 | 0.657 | 0.648 | 0.657 | 0.648 |
| 2.2 | 5 | 1 | 1 | 0.900 | 0.881 | 0.100 | 0.119 |
| 2.1 | 3 | 1 | 1 | 0.878 | 0.853 | 0.122 | 0.147 |
| 4.2 | 3 | 1 | 1 | 0.718 | 0.710 | 0.718 | 0.710 |
| 1.3 | 1 | 1 | 1 | 0.958 | 0.959 | 0.042 | 0.041 |

