# OpenReview forum: "Unpacking the Suitcase of Semantic Similarity"
_ICLR.cc/2026/Conference — ICLR 2026 Conference Withdrawn Submission_

### Official Review · Reviewer_v8qW · 2025-10-20

**Soundness:** 2
**Presentation:** 1
**Contribution:** 2
**Rating:** 2
**Confidence:** 4

**Summary:**

This paper considers the issue of semantic similarity used for retrieving documents for RAG purposes. The concept itself is overloaded and the paper proposes a logical entailment based analytical framework to create a benchmark dataset and an evaluation of 15 text embedding models using it. They find propose contextual rather than semantic similarity as what is being measured by existing text embedding models.

**Strengths:**

- The concepts of semantic similarity, semantic embedding and entailment are often used ambiguously and their effect on the retrieval step in RAG is not well understood. In this regard, the paper is aiming to evaluate a missing gap in RAG, which is commendable.
- A propositions dataset has been created, which could be potentially useful for other semantic similarity evaluation tasks outside RAG.
- In the experimental evaluations the authors have selected some of the SoTA text embedding models, which could be of interest to a broader audience given their wide applications/popularity.

**Weaknesses:**

- Although the paper claims to consider RAG as the main task, I did not find any RAG specific evaluations in the paper. There are different RAG methods and it is not clear how the observed trends would generalise across different RAG methods and not only across different text embedding models.
- There is much prior work on lexical entailment (see for example SelEval task series on [lexical entailment](https://aclanthology.org/2020.semeval-1.2/)). The connection to this line of work and the novelty of the paper should have been discussed.
- The difference between entailment and similarity has been studied previously (e.g. [Yokote et al.](https://ojs.aaai.org/index.php/AAAI/article/view/8348) ) and their non-equivalence has been highlighted. Although that work is not on LLM-based text embeddings, I think the conclusion (similarity is not entailment) is consistent with this paper and should be discussed as a prior related work.
- The conclusions of made in the paper are not strictly relevant to RAG (it does not matter for the practitioners of RAG whether the relevance between the documents and queries are measured using semantic or contextual similarity).

**Questions:**

- Although you have evaluated different text embedding methods, I did not see any evaluations involving different RAG methods. Why is this?
- What are the practical implications of this work to the practitioners of RAG?

---

### Official Review · Reviewer_NDkQ · 2025-10-27

**Soundness:** 3
**Presentation:** 3
**Contribution:** 2
**Rating:** 2
**Confidence:** 4

**Summary:**

The paper „Unpacking the Suitcase of Semantic Similarity” looks at the relationship between the commonly used cosine similarity and different, more specific semantic concepts related to the entailment of statements, i.e., entailment, contradictions, and independence. The authors train linear projections into a new embedding space from a model’s output embedding, such that the cosine similarity in this embedding space is aligned with the semantic entailment, i.e., correlates well with the conceptual expectation from the entailment operation (-1 for contradictions, 0 for independence, +1 for entailment). This is tested with multiple LLMs and the results show that such a linear projection can indeed yield an embedding, that is more closely aligned to this concept.

**Strengths:**

I completely agree that a lot of people misunderstand what semantic similarity measures and that they misuse this often when designing simplistic RAG approaches. I also believe that a more nuanced approach towards semantic similarity is required that considers different pragmatic aspects of the measure. The paper provides many details about the experiment setup and a nice derivation of a formal model for a semantic similarity variant for entailment similarity.

**Weaknesses:**

1) The conclusion, that the semantic similarity that is computed by encoder-only models should rather be called “contextual similarity” is hardly surprising. Indeed, this is commonly used when talking about LLM embeddings with more precision, e.g., when explaining the difference between word2vec to BERT. This is also a key question I would raise: how is this new? This is rather about precision of language, where we lazily just say semantic similarity instead of being more precise and talk about contextual similarity. Is there more to this that I am missing? Currently, I find this a bit weak for a core conclusion of the paper.
2) I wonder how the projection that is learned is different from an entailment classifier that uses cosine similarity as activation function instead of SoftMax. How the measurements here compare to a normal SoftMax classifier that is trained on the embeddings is unclear to me. Moreover, since the semantic similarity should actually got to +1/-1 for entailment/contradiction, wouldn’t an evaluation of the actual distribution not provide relevant insights into whether the projected embeddings lead to similarity measures that work as expected for entailment (see, e.g., https://openreview.net/forum?id=bfsNmgN5je)?
3) The data creation described in Section 3.1 relies a lot on LLMs, with sometimes a human in the loop to resolve inconsistencies. What I am missing is data for the reliability of this process, e.g., how well the human agreed with LLM annotations, if everything was checked or if the “remaining inconsistencies” were somehow identified and the human just looked at a subset of the data, and to which degree the human acted on the final LLM-based consistency checked and why/why not this was the case. Basically, I am missing all the information I would expect if there would be multiple human annotators.
4) All references are broken (missing brackets).

**Questions:**

1) What is new about the finding that cosine similarity between embeddings measures the similarity of the whole context of two statements and not entailement?
2) To which degree does the experiment yield new insights about the quality of embeddings with respect to entailment that cannot be obtained by just training a classifier for entailment, as is part of many benchmarks, including GLUE and SuperGLUE?

---

### Official Review · Reviewer_6n2m · 2025-10-30

**Soundness:** 2
**Presentation:** 2
**Contribution:** 2
**Rating:** 2
**Confidence:** 4

**Summary:**

This paper identifies a disconnect between the expectation of retrieving semantically highly relevant text and the kinds of information that text embeddings actually represent. It aims to unpack "semantic similarity" into empirically distinguishable components and explore how they factor into the cosine similarity of text embeddings. To investigate this issue, a benchmark dataset of concepts and propositions with quantitatively characterized semantic entailment relationships is created. The paper suggests using "contextual similarity" rather than "semantic similarity" when referring to cosine similarity estimates from text embedding models.

**Strengths:**

- The paper proposes a new dataset for evaluating semantic entailment relationships.
- The suggestion to use the term "contextual similarity" instead of the broad and vague "semantic similarity" for retrieval tasks is thought-provoking and could help clarify discussions in the field.
- The analysis of disentangling contextual and entailment similarity is interesting and provides valuable insights into what embedding models actually capture.

**Weaknesses:**

- Embedding is a fundamental concept applicable to broad applications. The paper would benefit from explicitly constraining its scope to information retrieval to differentiate from semantic embeddings used in other contexts. It is unclear whether the authors are aware of the well-established tasks of Semantic Textual Similarity (STS) and Natural Language Inference (NLI), which are highly relevant to the proposed ideas. When discussing STS tasks specifically, the naming of "semantic similarity" is already well-established and widely accepted in the community.
- The paper lacks a thorough discussion of how the proposed benchmark differs from widely-used datasets like STS-B and NLI benchmarks (e.g., SNLI, MultiNLI). A comparative analysis would strengthen the contribution and clarify the unique value of the proposed dataset.
- The prompts used for embedding evaluation are not provided. This is critical because modern embedding models (as evaluated in MTEB) support task-specific prompts or instructions to optimize performance for different tasks (STS, Retrieval, Clustering, etc.). For this paper's evaluation to be fair and reproducible, it should provide prompts or instructions that were used.
- The paper would be strengthened by including experiments on established STS and NLI datasets to demonstrate how the insights from the proposed benchmark generalize to existing benchmarks.
- There is insufficient discussion about the quality control process for the synthetic data generation. Given that the benchmark is a core contribution, more details on validation and quality assurance are needed.

**Questions:**

**Questions:**
- Were you specifying the proper task-specific prompts when evaluating embedding models on the proposed dataset? In the context of this paper, STS-specific prompts should be used.
- Was there a manual quality checking process for the synthetic data? What percentage of the data was manually validated, and what were the quality metrics?

**Suggestions:**
- It would strengthen the paper to include a detailed comparison table between the proposed dataset and existing benchmarks (STS-B, NLI datasets), and highlight their differences.
- It would be better to conduct experiments on STS-B and NLI datasets to validate whether the insights from your benchmark generalize to these established benchmarks.


**Typo and layout**:
- Center Table 1
- L53: "suitcase" -> ``suitcase''

---

### Official Review · Reviewer_y8zr · 2025-11-04

**Soundness:** 2
**Presentation:** 1
**Contribution:** 1
**Rating:** 2
**Confidence:** 4

**Summary:**

This paper aims to decompose the concept of "semantic similarity" in text embeddings by distinguishing between contextual similarity and semantic entailment. The authors develop analytic expressions for semantic entailment similarity at concept, predicate, and proposition levels based on a logical framework of conceptual semantics. They create benchmark datasets with ground truth entailment relationships, train linear projections from 15 embedding models to predict semantic entailment, and analyze which models better capture entailment versus contextual information. The authors conclude that current embeddings primarily capture contextual rather than entailment similarity and propose using "contextual fingerprint" instead of "semantic embedding."

**Strengths:**

- The distinction between different types of semantic similarity in embeddings is theoretically interesting and practically relevant for retrieval applications.
- The creation of controlled datasets with well-defined entailment relationships provides a principled evaluation framework.
- Comprehensive model evaluation: Testing 15 state-of-the-art embedding models provides reasonable empirical coverage.

**Weaknesses:**

- Vague and poorly explained methodology: The core theoretical framework is in appendices, making the main paper hard to follow, Key concepts like "semantic entailment similarity" are never clearly illustrated with concrete examples in the main text, The connection between the formal definitions and practical implementation is unclear

- The main result that embeddings capture "contextual" rather than "entailment" similarity is unsurprising given how embeddings are trained. The conclusions don't provide actionable insights for improving embedding models or retrieval systems

- Unclear practical relevance: The proposed terminology changes ("contextual fingerprint" vs "semantic embedding") seem pedantic, The framework requires manually constructed ontologies and predefined entailment relationships, limiting scalability, No clear path from analysis to improved systems

Overall, while the paper tackles an interesting theoretical question about the nature of similarity in embeddings, it suffers from overly complex presentation, limited empirical validation, and unclear practical implications. The core finding that embeddings capture contextual rather than logical entailment is neither surprising nor actionable. The paper would benefit from clearer exposition, more robust experimental validation, and concrete recommendations for improving embedding-based systems.

**Questions:**

How would this analysis help practitioners improve their embedding-based retrieval systems? What specific recommendations emerge beyond terminology changes?

The approach requires manually curated ontologies and predefined entailment relationships. How could this scale to real-world applications with diverse vocabularies?

Most embedding models have 768+ dimensions, yet they are projected down to as low as 50 dimensions. This seems counterintuitive - why would reducing dimensionality help capture more semantic information?

What's the point of character/token similarity measures? The paper mentions these as "symbolic similarity endpoints" but never clearly explains: How they relate to the main research question? What insights they provide beyond surface-level string matching?

Why linear projections at all? Why not train the embedding models themselves on entailment tasks?

---

### Note · Authors · 2025-11-18

I have read and agree with the venue's withdrawal policy on behalf of myself and my co-authors.